# Thromboinflammation Model-on-A-Chip by Whole Blood Microfluidics on Fixed Human Endothelium

**DOI:** 10.3390/diagnostics11020203

**Published:** 2021-01-29

**Authors:** Alexander Dupuy, Lejla Hagimola, Neil S. A. Mgaieth, Callum B. Houlahan, Renee E. Preketes-Tardiani, Paul R. Coleman, Freda H. Passam

**Affiliations:** 1Heart Research Institute, University of Sydney, Sydney, NSW 2042, Australia; alexander.dupuy@hri.org.au (A.D.); lejla.hagimola@hri.org.au (L.H.); nmga6623@uni.sydney.edu.au (N.S.A.M.); callum.houlahan@hri.org.au (C.B.H.); rpre3982@uni.sydney.edu.au (R.E.P.-T.); p.coleman@centenary.org.au (P.R.C.); 2Centenary Institute, University of Sydney, Sydney, NSW 2050, Australia

**Keywords:** neutrophil, thromboinflammation, microfluidics, endothelium, platelet

## Abstract

Microfluidic devices have an established role in the study of platelets and coagulation factors in thrombosis, with potential diagnostic applications. However, few microfluidic devices have assessed the contribution of neutrophils to thrombus formation, despite increasing knowledge of neutrophils’ importance in cardiovascular thrombosis. We describe a thromboinflammation model which uses straight channels, lined with fixed human umbilical vein endothelial cells, after treatment with tumour necrosis factor-alpha. Re-calcified whole blood is perfused over the endothelium at venous and arterial shear rate. Neutrophil adhesion, platelet and fibrin thrombus formation, is measured over time by the addition of fluorescent antibodies to a whole blood sample. Fixed endothelium retains surface expression of adhesion molecules ICAM-1 and E-Selectin. Neutrophils adhere preferentially to platelet thrombi on the endothelium. Inhibitors of neutrophil adhesion and anti-inflammatory agents, such as isoquercetin, decrease neutrophil adhesion. Our model offers the advantage of the use of (1) fixed endothelium, (2) whole blood, instead of isolated neutrophils, and (3) a small amount of blood (1 mL). The characteristics of this thromboinflammation model provide the potential for further development for drug screening and point-of-care applications.

## 1. Introduction

Vascular inflammation triggers thrombosis in cardiovascular disease such as heart attack and stroke [1,2]. Neutrophils are critical to the development of thromboinflammation [3]; neutrophil adhesion to inflamed endothelium potentiates atherosclerosis [4], whereas neutrophil–platelet aggregates cause microvascular infarction in ischemia-reperfusion [5]. Drugs inhibiting neutrophil adhesion to endothelium and platelets, such as a recent inhibitory antibody of integrin Mac-1, have shown potential in reducing inflammation without compromising immunity [6]. There is a need to incorporate the study of neutrophils in diagnostics for the evaluation of cardiovascular risk or for monitoring of anti-inflammatory treatments.

Diagnostics in cardiovascular inflammation rely on direct visualisation of the vasculature (i.e., coronary angiogram or computerised tomography) and the measurement of surrogate markers of inflammation in the serum (i.e., high sensitivity C reactive protein) [7]. These investigations provide information on the existing damage but do not report on the function of an individual’s blood cells in relation to the vasculature. Furthermore, these tests do not account for interactions that occur between blood cells and the endothelium under conditions of flowing blood [8,9]. The introduction of endothelialised microfluidic devices to measure interactions of circulating cells with the endothelium, offers a significant tool to study thromboinflammation and to test the efficacy of drugs targeting neutrophil–endothelial or neutrophil–platelet interaction.

Neutrophil–endothelial interaction is mediated by the binding of neutrophil cell surface receptors, including P-selectin glycoprotein ligand 1 (PSGL-1), macrophage antigen 1 (Mac-1) and lymphocyte function-associated receptor 1 (LFA-1), to endothelial cell adhesion molecules, such as selectins, intercellular adhesion molecule 1 (ICAM-1), and vascular cell adhesion molecule 1 (VCAM-1) [10,11]. Neutrophil–platelet interaction is mediated by the binding of PSGL-1 and Mac-1 to platelet P-selectin and glycoprotein Ib alpha (gpIbα) respectively [12]. Protein disulphide isomerase (PDI), a circulating vascular thiol isomerase, promotes neutrophil–platelet interaction by enhancing Mac-1–gpIbα binding [13]. PDI is a promising target for the prevention of thromboinflammation [14,15,16] which can be studied in microfluidic systems incorporating endothelial cells and whole blood.

Microfluidic chips involve the manipulation of small sample volumes through a series of microchannels to perform biological assays [8,17]. Endothelialised microfluidic chips use endothelial cell monolayers to line the microchannels and are connected to equipment that can generate controlled flow, such as syringe or vacuum pumps. Microfluidic chips enable cellular interactions to be studied in a system which simulates the forces and structures found within the vascular system [18]. They have been useful in detecting the pro-thrombotic tendency of certain drugs which was not identified in pre-clinical animal studies [19]. Perfusion of whole blood in endothelialised chips has been used predominantly to study platelet adhesion and thrombus formation [20]. The study of whole blood neutrophil adhesion in the context of cardiovascular thrombosis is limited with chips utilising isolated neutrophils or cell lines (e.g., THP-1) [21,22]. A summary of the published endothelialised microfluidic chips assessing thromboinflammation to date is provided in Table 1.

In this study, we evaluate the application of a fixed-endothelial whole-blood microfluidic model for the study of neutrophil involvement in thromboinflammation. We further test the application of this chip for the evaluation of inhibitors of neutrophil adhesion and thrombus formation.

## 2. Materials and Methods

### 2.1. Reagents and Antibodies

The antibodies used in this study are shown in Table 2. The anti-fibrin antibody clone 59D8 and anti-P-selectin antibody G1 were from Gary Matsueda, Harvard University, USA [37] and Roger McEver, Oklahoma Medical Research Foundation, USA [38] respectively. Where stated, antibodies were labelled with the fluorescent labelling kits. Human fibronectin was purchased from Haematologic Technologies, Essex Junction, VT, USA. Collagen was from Helena laboratories, Beaumont, TX, USA. Polydimethylsiloxane kits (Sylgard 184) were from Dow Corning, Pennant Hills, NSW, Australia. Recombinant TNF-α was from R&D systems, Minneapolis, MN, USA. Phalloidin-AlexaFluor 568 was from Invitrogen, Carlsbad, CA, USA. Hoechst 33342 was from Invitrogen. Protamine sulphate was from Sanofi-Aventis, France. Isoquercetin (Quercetin 3-glucoside) and bovine serum albumin (BSA) were from Sigma-Aldrich, St Louis, MO, USA.

### 2.2. PDMS Device Fabrication and Endothelialisation

PDMS microfluidic devices were fabricated as previously described [39]. To prevent channel leakage, PDMS casts were assembled by plasma bonding onto glass coverslips. The coverslip and PDMS cast were exposed to oxygen plasma for 1 min, and PDMS casts were placed channel side down onto coverslips (Figure 1). The assembled devices were then heated at 60 °C for 15 min. The devices were left to equilibrate to room temperature before coating the channel with fibronectin (100 µg/mL) or collagen (100 µg/mL) and then incubated overnight at 4 °C.

Human umbilical vein endothelial cells (HUVECs) were obtained from umbilical cords by collagenase treatment [40], after donor informed consent, under Ethics approval by the Sydney Local Health District Human Ethics Committee, X16-0225. HUVECs were isolated and maintained in culture in MesoEndo growth medium (Lonza, Basel, Switzerland) in 5% CO_2_ incubator. HUVECs were cultured to 90% confluency. The culture medium was then replaced with fresh media, with or without TNF-α (10 ng/mL), and HUVECs were left to incubate for 16 h. Cells were detached from culture flasks using Trypsin/EDTA and resuspended at 10–20 × 10^6^ cells/mL and 10 uL was perfused through the microfluidic channel. Cells were left to settle and spread on the fibronectin channel for 1 h, washed with PBS, then fixed with 4% paraformaldehyde (PFA) for 10 min. The PFA was then flushed from the channels with PBS. Fixed endothelialised biochips were stored at 4 °C for up to 1 week before use.

HUVECs’ confluency and surface expression of adhesion proteins, within the devices, were evaluated by fluorescent staining. Non-permeabilized HUVECs were blocked with 2% (*w*/*v*) BSA for 30 min at room temperature followed by staining with anti-ICAM-1-APC (10 µg/mL), anti-von Willebrand factor (vWF)-Alexa 594 (2 µg/mL) for 30 min at room temperature or E-Selectin (CD62E) (10 µg/mL) for 2 h at room temperature, followed by staining with anti-rat Ig-FITC (20 μg/mL) for 2 h at room temperature. In other experiments, HUVECs were permeabilised by perfusing 0.5% (*v*/*v*) Triton X100/PBS containing 5% (*w*/*v*) BSA, through the channel and incubating at room temperature for 30 min. The devices were then washed with PBS and blocked with 2% (*w*/*v*) BSA for 30 min at room temperature. The endothelialised channels were then stained with Phalloidin-AlexaFluor 568, ICAM-1-APC, or anti-VE-Cadherin (0.5 ug/mL) antibody, followed by incubation with secondary goat anti-rabbit IgG-AlexaFluor 488 (2 µg/mL). HUVECs were stained with Hoechst 33,342 (3 µg/mL) for 15 min, washed with PBS and then imaged.

### 2.3. Microfluidics Assay

Venous blood was collected from healthy human volunteers in accordance with the Human Research Ethics Committee of the University of Sydney (2014/244) and the declaration of Helsinki. Whole blood was collected using a 21 G needle into either 3.2% sodium citrate or lithium heparin spray-coated vacutainers (Becton Dickinson) providing a heparin concentration of 17 International Units (IU)/mL blood. Whole blood was used within 2 h of collection.

Whole blood was stained for platelets using anti-CD41 antibody (0.5 µg/mL) or anti-CD42a antibody (20 µL per 1 mL of whole blood), fibrin using anti-fibrin antibody (0.5 µg/mL), and neutrophils using anti-CD66b antibody (0.25 µg/mL) or anti-CD18 antibody (2 µg/mL). For some experiments, whole blood was also treated with inhibitory anti-CD11b (10 µg/mL) [41], inhibitory anti-P-selectin (CD62P) (10 µg/ml) [38] or isoquercetin (100 µM) [42]. Reagents were incubated for 10 min at room temperature. Heparin in whole blood was reversed by addition of protamine sulphate (1 mg protamine per 100 IU heparin) immediately before being added to the inlet of the endothelialised biochip. Aliquots of citrated whole blood were re-calcified by adding CaCl_2_ to a final concentration of 10 mM immediately before being added to the inlet of the endothelialised biochips. Whole blood was perfused at a “venous” shear rate of 100 s^−1^ and “arterial” shear rate of 700 s^−1^ [18,43].

### 2.4. Image Acquisition and Analysis

Images were acquired at 10 min on a Zeiss 880 with AiryScan confocal microscope. 5 × 3 tile-scans were imaged on a 20× objective (NA 0.8). After 15 min of whole blood perfusion, z-stacks were taken using a 40× water immersion objective (NA 1.2) at the beginning, middle, and end of the channel.

For high shear experiments, images were acquired after 4 min of whole blood perfusion and compared to 4 min of whole blood perfusion under low shear.

Surface area coverage of platelets and fibrin were quantified using maximum intensity projections generated from z-stacks. Images were thresholded on ImageJ (https://imagej.net/Fiji) to produce a binarized image, and the pixels above threshold were analysed to determine the area of coverage.

Neutrophil adhesion was quantified by counting the number of adherent neutrophils on z-stacks using ImageJ. An adherent neutrophil was defined as a neutrophil that moved no more than 1 cell diameter (~10 µm) within 5 s from the initial point of attachment [44].

### 2.5. Statistical Analysis

Significance testing was performed by one-way ANOVA with Dunnett’s post-hoc test or non-parametric *t*-test (Wilcoxon rank), where stated.

## 3. Results

### 3.1. Microfluidic Device Endothelialisation

To improve the lead time and simplicity of biochip production, we followed a modified protocol for endothelial cell priming and surface endothelialisation, which does not require an overnight cell culture perfusion system [21]. Immunostaining showed a confluent endothelial surface (Figure 2). Non-TNF-α-treated HUVECs do not express ICAM-1 or E-selectin, whereas TNF-α-treated HUVECs express both ICAM-1 and E-selectin. On the contrary, both non-TNF-α-treated HUVECs and TNF-α-treated HUVECs express the same amount of vWF (Figure 3).

### 3.2. Technical Considerations of Whole Blood Perfusion on Endothelialised Chip

#### 3.2.1. Effect of Anticoagulant on Thromboinflammation Chip

Anticoagulants commonly used for blood collection include citrate, EDTA and heparin. Citrate and EDTA chelate calcium ions which are necessary for cell–cell interaction and coagulation. Heparin inhibits factor Xa and thrombin, thus preventing coagulation. The coagulation potential of citrated or heparinized blood can be restored by recalcification or addition of protamine sulphate, respectively.

In our system, when citrated or heparinized whole blood was perfused through TNF-α-treated endothelial biochips, there were no thrombi formed (Figure 4A,C). However, fibrin, platelet and neutrophil deposition was observed in biochips perfused with recalcified citrated blood or by addition of protamine to heparinized blood (Figure 4B,D). We found that recalcification of citrated blood with 10 mM CaCl_2_ provided adequate time for consistent fibrin, platelet, and neutrophil adhesion to the endothelialised channel at 100 s^−1^ within the 15 min capture, without complete occlusion of the microfluidic channel inlet. Other groups recalcify with CaCl_2_ alone [24] or CaCl_2_ and MgCl_2_ [18]. It is advisable that the individual laboratory titrates the concentration of CaCl_2_ and MgCl_2_ to achieve fibrin, platelet and neutrophil adhesion according to experimental shear rates and timeframe.

The recalcification step is critical for unobstructed perfusion of whole blood on endothelialised channels. Various mixing devices have been developed, including the introduction of a herringbone mixer [45] or parallel sheath flow [46] for continuous recalcification. Despite the increase in complexity of the design, mixing elements may be important when longer captures are required.

#### 3.2.2. Effect of Endothelial Cell Substrate on Thromboinflammation Chip

Substrates commonly used for endothelial cell attachment to the biochip include fibronectin and collagen [18,30]. Fibrin and platelet adhesion were comparable when whole blood was perfused over endothelial cells attached to fibronectin versus collagen (Figure 5A–D), whereas neutrophil adhesion was increased on collagen versus fibronectin-coated substrate (Figure 5E).

#### 3.2.3. Effect of Storage on Thromboinflammation Chip

A benefit of using fixed endothelial chips is that these can be batch-made, stored for later use or shipped to laboratories which do not have facilities to assemble the endothelialised chips. We compared the adhesion of platelets, neutrophils on endothelialised chips one day versus seven days after fixation. These show comparable thromboinflammation (Figure 6).

#### 3.2.4. Effect of Shear Rate on Thromboinflammation Chip

A range of shear rates can be employed in the microfluidics device to simulate venous or arterial blood flow. The endothelialised chip can be used to support thromboinflammation at low and high shear (Figure 7A–E). At higher shear rate (700 s^−1^) neutrophils adhere faster than at low shear rate (100 s^−1^). At 4 min of perfusion nearly no neutrophils adhered at 100 s^−1^ versus an average of three neutrophils per field at 700 s^−1^ (Figure 7E).

### 3.3. Use of Endothelialised Biochips to Study Thromboinflammation

Inhibition of the function of Mac-1 (CD11b/CD18), a major integrin receptor on the surface of neutrophils and monocytes, inhibits thrombus formation [47]. To determine if this interaction can be observed in this model, recalcified whole blood was treated with an inhibitory concentration of the anti-CD11b antibody M1/70 and perfused through endothelial biochips.

In this biochip, we observed minor platelet adhesion and fibrin deposition after 15 min of whole blood perfusion on non-TNF-α treated HUVECs-on chip. However, TNF-α treated HUVECS-on chip had significantly higher platelet, fibrin, and neutrophil adhesion (Figure 8).

While Mac-1 inhibition had no impact on fibrin deposition, a significant decrease in neutrophil adhesion and platelet coverage was observed (Figure 8C,D). P-selectin inhibition significantly decreased neutrophil and platelet adhesion (Figure 8C,D). This demonstrates the utility of this endothelial model to study the effect of inhibitors of neutrophil–platelet interaction in reducing thromboinflammation.

To further evaluate the translational potential of the fixed endothelial model, we examined the effect of isoquercetin on thromboinflammation. PDI is known to promote platelet accumulation, fibrin formation, and neutrophil recruitment [14,48,49]. Isoquercetin is a flavonoid PDI inhibitor [42] which has recently shown efficacy in the prevention of cancer-associated thrombosis [16]. Incubation of whole blood with an inhibitory concentration of isoquercetin, resulted in a significant decrease in the accumulation of fibrin, platelets, and neutrophils on the biochip (Figure 8B–D).

## 4. Discussion

We have demonstrated the assembly of a microfluidic device lined with fixed human endothelial cells that provides a reproducible platform to study neutrophil–platelet–endothelial cell interactions and the effect of drugs to prevent thromboinflammation. There are a number of steps which simplify our model: (1) Endothelial cells are introduced into the channel and allowed to adhere without the need for long term perfusion, (2) endothelial cells are fixed and the chip can be stored in 4 °C for one week without loss of activity, and (3) small quantities of whole blood are used.

Previously similar microfluidic models have been designed to study thrombosis and vascular inflammation (summarized in Table 1). In particular, Jain, et al. produced a fixed endothelialised microfluidic chip to study platelet adhesion and fibrin formation [18]. Our model extends the use of fixed endothelium for the study of neutrophil involvement in thrombus formation on TNF-α treated endothelium. The use of a whole blood system provides a physiologically relevant model where all components of blood are present during thrombus formation. A limitation of our fixed endothelial chip is that it is not suitable for the study of the effect of vWF on thromboinflammation, as this requires live endothelial cells for active secretion of vWF under shear. Published methods, using purified platelets and neutrophils to study their interaction with vWF, have been included in Table 1 [33,34,50]. A modification of our chip including live endothelial cells, for the study of endothelial-derived vWF in thromboinflammation, is provided in Appendix A.

Our model shows utility in studying mechanisms of thromboinflammation and the effect of potential inhibitors with high spatial and temporal resolution. The inhibition of Mac-1 integrin by inhibitory antibody M1/70 decreased platelet and neutrophil adhesion. Mac-1 interacts with GP1bα on platelets and with P-selectin on endothelial cells and platelets [42,46]. Inhibition of Mac-1 by M1/70 antibody has previously been shown to inhibit neutrophil adhesion to P-selectin [51]. Despite the effectiveness of Mac-1 inhibitors (i.e., anti-CD18 antibodies) to improve thrombosis outcomes in animal models, they failed to improve outcomes in clinical trials [52]. More understanding of the role of Mac-1 and other cell surface receptors in thrombosis/inflammation can be achieved by studying the interactions of human blood and endothelial cells in microfluidic platforms.

Additionally, our model is suitable for the study of the antithrombotic effect of anti-inflammatory agents. Isoquercetin is a flavonoid that is consumed as a dietary supplement. Epidemiological studies have shown that increased flavonoid consumption is linked to a decreased incidence of coronary artery disease [53]. In a Phase II clinical trial, isoquercetin decreased the incidence of cancer-associated venous thromboembolism [15]. Isoquercetin inhibits PDI in vitro and in vivo; however, has additional effects as an antioxidant and anti-inflammatory agent [54]. Whole blood treated with isoquercetin inhibited platelet and neutrophil adhesion and fibrin formation in our chip, consistent with isoquercetin’s function in in vivo models [42,55].

While microfluidic models of thrombosis focus on studying platelet and fibrin kinetics, recent developments in these models have started to focus on neutrophil-thrombus interactions (Table 1). Neutrophil–platelet interactions promote thrombus formation, cardiovascular inflammation, and atherosclerosis by recruitment of monocytes and proteolysis [4]. Furthermore, neutrophils promote thrombosis by releasing neutrophil extracellular traps (NETs). NETs play an important role in the development of thrombosis [56], a process which can be studied in our microfluidic model [57].

Although using fixed endothelial chips is a step towards simplification in the application of microfluidic devices, further optimisation is required. A major disadvantage of endothelialised devices is the difficulty in production and storage due to the dependence on endothelial cell cultures for the provision of endothelial cells for the chips. Additionally, microfluidic chips require nanofabrication expertise and optimisation of fluidic parameters [20]. International groups such as the International Society of Thrombosis and Hemostasis Standardization Subcommittee have a dedicated taskforce for the development of guidelines and the standardization of microfluidic assays [58]. These international taskforces may facilitate the optimisation of these assays for entry into diagnostics and point of care applications.

## Figures and Tables

**Figure 1 diagnostics-11-00203-f001:**
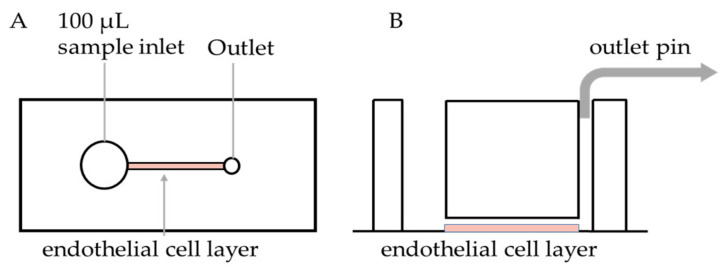
Design of endothelialised microfluidic chip. (**A**) Top and (**B**) side views of the Polydimethylsiloxane (PDMS) chip. Each chip consists of an inlet well, straight channel and an outlet pin connected to the pump which perfuses the inlet fluid at pre-determined shear rates. An endothelial cell suspension is added to the well, perfused over the channel and allowed to create a monolayer on the surface.

**Figure 2 diagnostics-11-00203-f002:**
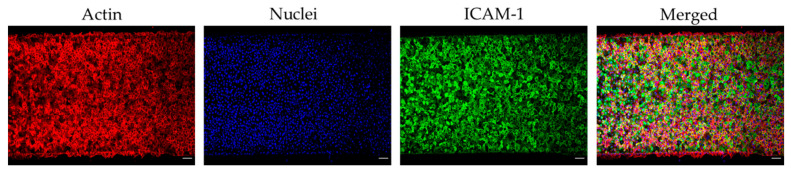
Microchannel coated with tumour necrosis factor alpha (TNF-α) treated human umbilical vein endothelial cells (HUVECs). Confocal micrographs showing tile-scans of endothelialised devices stained for actin (red), nuclei (blue), and intercellular adhesion molecule 1 (ICAM-1) (green). Scale bars represent 100 µm.

**Figure 3 diagnostics-11-00203-f003:**
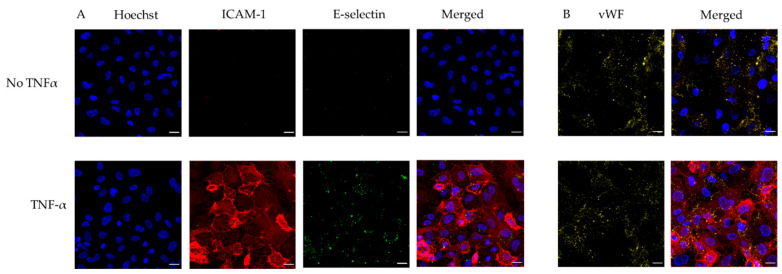
Expression of surface adhesion proteins on human umbilical vein endothelial cells (HUVECs) in the endothelialised chips. (**A**) Confocal micrographs showing maximum intensity projections of z-stacks stained for nuclei (blue), intercellular adhesion molecule 1 (ICAM-1) (red), E-selectin (green) and (**B**) von Willebrand factor (vWF) (yellow) of non-treated (no tumour necrosis factor alpha (“No TNF-α”) and TNF-α treated (“TNF-α”) HUVECs-on chip. Scale bars represent 20 µm.

**Figure 4 diagnostics-11-00203-f004:**
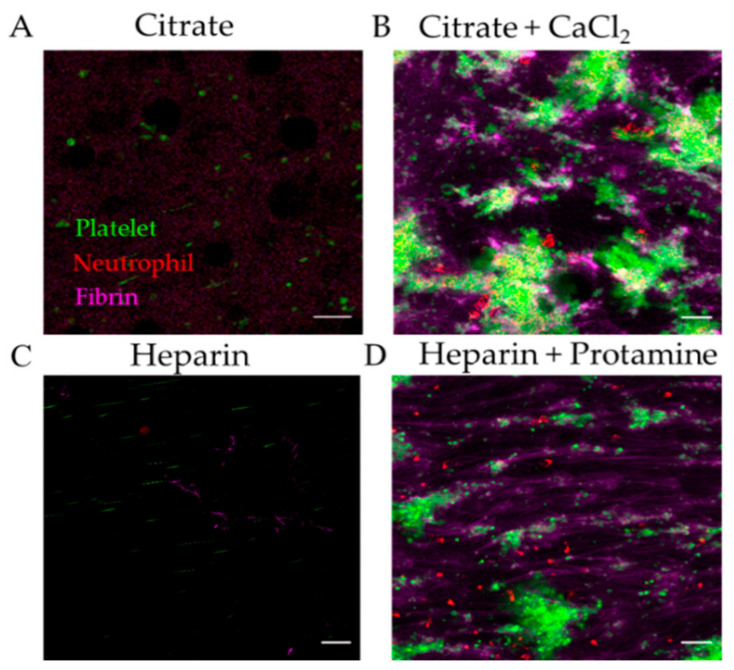
Effect of anti-coagulant on performance of the thromboinflammation chip. Representative images of fibrin (magenta), platelet (green), and neutrophil (red) adhesion on endothelialised chip (**A**) without or **(B**) with recalcification of citrated blood with 10 mM CaCl_2_; (**C**) without or (**D**) with addition of protamine to heparinized blood. Scale bar represents 20 µm.

**Figure 5 diagnostics-11-00203-f005:**
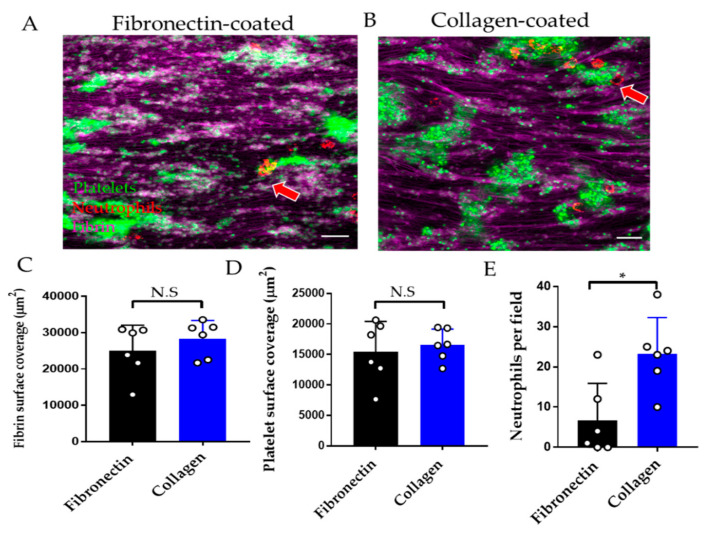
Effect of endothelial substrate on performance of the thromboinflammation chip. Representative images of fibrin (magenta), platelet (green), and neutrophil (red) adhesion on endothelialised chip on (**A**) fibronectin or (**B**) collagen substrate. Red arrows denote the contact of neutrophils with platelet aggregates. (**C**) Fibrin, (**D**) platelet surface coverage, and (**E**) neutrophils per ×40 field on fibronectin coated (black columns) versus collagen-coated (blue columns) endothelialised chips. Data is mean ± SD of six fields of view from *n* = 3 separate donors. * *p* < 0.05; N.S = non-significant by two-tailed, paired non-parametric *t*-test (Wilcoxon).

**Figure 6 diagnostics-11-00203-f006:**
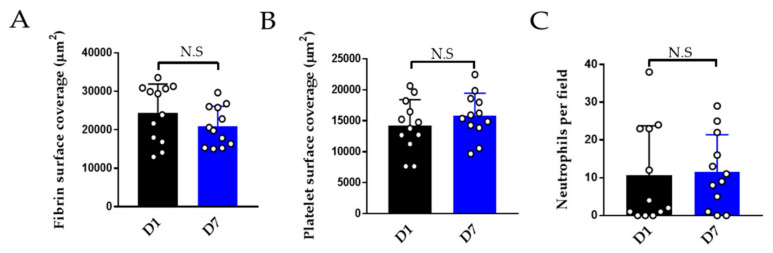
Effect of storage on performance of the thromboinflammation chip. Whole blood was perfused on endothelialised chips stored for one day versus seven days after fixation. (**A**). Fibrin (**B**) platelet surface coverage and (**C**) neutrophils per ×40 field on one-day-old (black columns) versus seven-day-old chips (blue columns) from HUVECs from the same cord (four cords in total). Data is mean ± SD of 12 fields of view from *n* = 4 separate donors. N.S = non-significant by two-tailed, non-parametric *t*-test (Wilcoxon).

**Figure 7 diagnostics-11-00203-f007:**
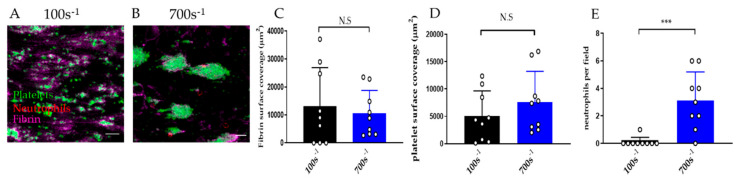
Effect of shear rate on performance of the thromboinflammation chip. Whole blood was perfused on endothelialised chips for 4 min at low (100 s^−1^) and high (700 s^−1^) shear. Representative images of fibrin (magenta), platelet (green), and neutrophil (red) adhesion on endothelialised chip after perfusion of recalcified citrated blood, at (**A**) low (100 s^−1^) versus (**B**) high (700 s^−1^) shear. (**C**) Fibrin and (**D**) platelet surface coverage and (**E**) neutrophils per 40× field at 4 min after perfusion at 100 s^−1^ (black columns) and 700 s^−1^ shear (blue columns). Data is mean ± SD of nine fields of view from *n* = 3 separate donors. *** *p* < 0.001, N.S = non-significant by two-tailed, non-parametric *t*-test (Wilcoxon).

**Figure 8 diagnostics-11-00203-f008:**
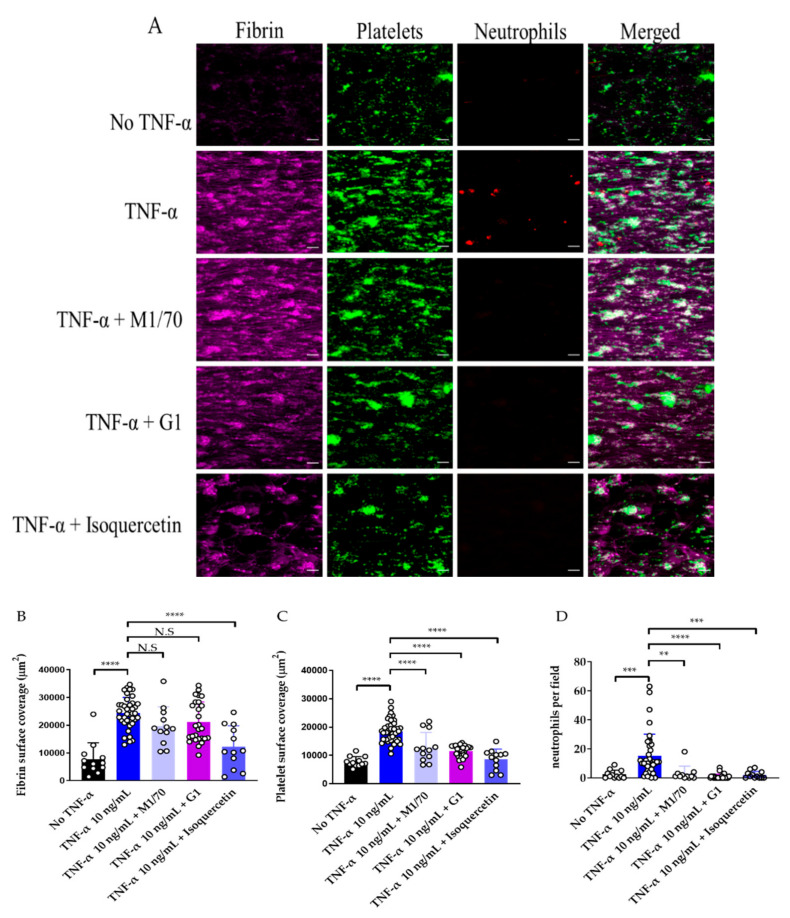
Effect of anti-inflammatory agents on thromboinflammation. (**A**). Representative images of fibrin (magenta), platelets (green), and neutrophils (red) on endothelialized chips. Endothelial cells were treated with tumour necrosis factor alpha (“TNF-α”) or without TNF-α (“No- TNF-α”). Whole blood was incubated with M1/70 (inhibitory antibody to Mac-1), G1 (inhibitory antibody to P-selectin) and isoquercetin. Scale bar represents 20 µm. (**B**). Fibrin, (**C**). platelet surface coverage and (**D**). neutrophils per x40 field in No TNF-α (black columns), TNF-α (dark blue columns), TNF-α in the presence of M1/0 (violet columns), TNF-α in the presence of G1 (magenta columns) and TNF-α in the presence of isoquercetin (light blue columns). Data represents mean ± SD of 3-9 fields of view per donor, n = 4–6. ** *p* < 0.01; *** *p* < 0.001; **** *p* < 0.0001, N.S = non-significant by one-way ANOVA with Dunnett’s post-hoc multiple comparison..

**Table 1 diagnostics-11-00203-t001:** Microfluidic endothelialised models used in basic and translational research for the study of vascular thromboinflammation.

Microfluidic Device Description	Endothelial Treatment	Perfusion Sample	Function Measured	Clinical/Diagnostics Potential	Reference
			**Hemostasis**		
HUVEC-lined straight channel	Mechanical injury by pneumatic valve	Re-calcified citrated whole blood	Hemostasis	Detection of hemostatic defects (hemophilia) and screening of anticoagulants	[23]
			**Thrombosis**		
HUVEC-lined straight channel	TNF-α and fixation	Re-calcified citrated whole blood	Platelet adhesion and fibrin generation	Long-term storage of chips for study of thrombosis	[18]
HUVEC-lined flow chambers	TNF-α	Re-calcified citrated whole blood	Fibrin formation, platelet and neutrophil adhesion	Testing anti-thrombotic potential of single-chain antibody fragment	[24]
HUVEC-lined straight channel	-	Re-calcified citrated whole blood	Fibrin formation	Study of complement-driven thrombosis	[25]
BOEC-lined straight channel	TNF-α	Re-calcified citrated whole blood	Platelet adhesion and fibrin generation	Non-invasive assessment of endothelial function and thrombus formation in diabetes	[26]
HUVEC-lined 3D channel with stenoses	-	Re-calcified citrated whole blood	Platelet adhesion and thrombus formation	Employing medical CTA imaging data to produce patient-specific microfluidic chips	[27]
HMVEC-lined straight channel	-	Re-calcified citrated whole blood	Fibrin formation	Study of endothelial function and thrombus formation in diabetes	[28]
			**Thromboinflammation**		
HAECS-lined curved channel with gradient shear	-	Isolated neutrophils, and monocytes	Neutrophil and monocyte adhesion	Study of the effect of shear stress on vascular inflammation	[29]
HUVEC, HLMVEC-lined 3D branching microchannels	TNF-α, STX2	Heparin, EDTA whole blood	Platelet, neutrophil and monocyte adhesion	Study of microvascular occlusion in sickle cell disease and HUS and effect of medications	[30]
HUVEC-lined 3D microvascular network	TNF-α	Isolated neutrophils	Neutrophil rolling, adhesion, migration	Study of vascular inflammation	[22]
HUVEC-lined straight channel	TNF-α	Re-calcified whole blood	NETosis	Study of NETosis and thrombosis in patient groups (HIT)	[31]
HUVEC-lined 3D channel with stenosis	TNF-α	Citrated whole blood	Platelet and neutrophil adhesion	Study of antithrombotic potential of aspirin and metformin	[32]
BOEC-lined straight channel	PMA, histone	Isolated platelets	Platelet adhesion to vWF strings	Study of thromboinflammation	[33]
HUVEC-lined parallel flow chamber	histamine	Isolated platelets, reconsitituted blood	Platelet and neutrophil adhesion to vWF	Study of neutrophil involvement in thrombosis under high shear	[34]
			**Endothelial function**		
HUVEC, HUASMC, HBVPC-seeded multi-channel network	PMA, Media containing angiogenic growth factors	Citrated whole blood	Angiogenesis, vessel permeability, thrombosis	Study of angiogenesis and microvascular thrombosis in CV medicine	[35]
Neuron, astrocytes and HUVEC-lined 3D multi-channel	-	Fluorescent dextran	Blood–brain barrier	Screening of drugs for blood–brain barrier permeability	[36]

HUVEC: Human umbilical vein endothelial cells; TNF-α: Tumour necrosis factor alpha; BOEC: Blood outgrowth endothelial cells; CTA: Coronary computed tomography angiography; HMVEC: Primary human cardiac microvascular endothelial cells; HAEC: Human aortic endothelial cells; STX2: Shiga toxin type 2; HUS: Hemolytic uremic syndrome; NETosis: The formation of neutrophil extracellular traps; HIT: Heparin induced thrombocytopenia; CV: Cardiovascular; HBVPCs: Human brain vascular pericytes; HUASMCs: Human umbilical arterial smooth muscle cells; HLMVEC: Human lung microvascular endothelial cells; PMA: Phorbol-12-myristate-13-acetate; vWF: von Willebrand factor.

**Table 2 diagnostics-11-00203-t002:** Antibodies used for the study of thromboinflammation in this study.

Target Antigen	Clone	Supplier	Fluorescent Label
CD42a	ALMA.16	Becton Dickinson, Franklin Lakes, NJ, USA	PE
CD18	H52	Developmental Studies Hybridoma Bank, university of Iowa, USA	AlexaFluor 488, labelled using protein labelling kit A10235 (Invitrogen)
CD66b	80H3	Beckman Coulter, Brea, CA, USA	APC
Fibrin	59D8	-	AlexaFluor 594, labelled using protein labelling kit A10239 (Invitrogen)
CD41a	P2	Beckman Coulter	FITC
CD11b	M1/70	Invitrogen	-
ICAM-1	HA58	Invitrogen	APC
VE-Cadherin	D87F2	Cell Signalling Technologies, Danvers, MA, USA	-
CD62P	G1	-	-
CD62E	UZ4	Thermofisher	-
vWF	rabbit polyclonal	Dako	AlexaFluor 594, labelled using protein labelling kit A10239 (Invitrogen)
Rabbit IgG	N/A	Invitrogen	AlexaFluor 488
Rat Ig	N/A	Becton Dickinson	FITC

## Data Availability

Data is contained within the article.

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
