# Peer review of "Thromboinflammation Model-on-A-Chip by Whole Blood Microfluidics on Fixed Human Endothelium"

_diagnostics, 2021, doi:10.3390/diagnostics11020203_

Round 1
Reviewer 1 Report
Dupuy et al reported a novel thromboinflammation model-on-a-chip that was proposed to use to study one of hot topics in thrombosis field. The authors provide a good overview about similar models alongside the characterization of their own model. This study has a merit, but I have several concerns.
In the method section, the authors indicate that citrate anticoagulated blood was recalcified by addition 10 mM final concertation of CaCl2 without MgCl2. Given that sodium citrate can also chelate magnesium, the blood should be also supplemented with that in order to provide a more physiological condition. Furthermore, this recalcification was performed by addition the CaCl2 directly to whole blood which can cause a rapid clotting in case of the pre-activation of the blood. Have the authors encountered any problems with blood clotting before the blood entering the channel or have flowing clots been visible during perfusion?
The authors used fibrin antibody to detect fibrin formation during flow. Based on the images presented in figure 5, the control condition and the TNF-alpha+Isoquercetin conditions make me wonder about the specificity of the fibrin antibody. Can this antibody recognize fibrinogen as well?
Furthermore, the formed fibrin looks different between TNF-alpha and TNF-alpha+M1/70 treated samples. In the latter condition the fibrin fibers are aligned with the flow, whereas the TNF-alpha treatment alone does not show it that clearly. Are these fibrins originated from plasma or platelets?
Figure 6 has to be revised. The authors should either indicate the various panels separately or provide a proper labeling of the y axes. Furthermore Figure 5 and 6 could be merged as those belong together.
It would be easier to compare the various images with the same scales.
The authors state in the discussion that their model can be stored for 1 week without loss of activity, however there is no data presented about that. Therefore, the authors should include additional data proving that.
This model was proposed to be a good approach in studying thromboinflammatory processes but was only tested under low shear condition based on the manuscript. Thromboinflammation can occur under higher shear condition as well, so could be this method easily modified for such a condition?
Reviewer 2 Report
The manuscript by Dupuy et al. describes an assay that could be used to study the formation of platelet-neutrophil mixed thrombi in inflamed vessels.
A similar device and assay has already been described by Jain et. Al Biomed Microdevices 2016 for the purpose of studying thrombosis. Dupuy and colleagues propose to re-purpose the device to study a very important aspect of thromboinflammation thus I support the publication of the manuscript, becasue the topic is to high relevance to scientist in the thrombosis field. However, I suggest the manuscript should undergo major revision to increase the value of the work. See detailed comments below:
- Since heparinized blood does not support neutrophil deposition and citrated blood needs to be recalcified to obtain neutrophil-rich thrombi it appears that fibrin deposition, but not necessarily platelets, support neutrophil binding. Thus it would not be possible to use this setup to study platelet-neutrophil interactions. To understand if this is the case the authors should show in this systems what is the effect of inhibitors of P-selectin and GPIb, the main adhesion receptors that support platelet-neutrophil interactions. Moreover, we suggest the authors to delve more deeply in this issue by testing a) different anticoagulants and different shear rates to achieve a condition where platelet adhesion but not fibrin deposition drive neutrophil binding and also test P-selectin and GPIb inhibitors in this new setup.
- Even though the authors have based their device on Jain et al. they use much higher concentrations of TNF-alpha which are not physiologically relevant, for shorter time frames and they perfuse already stimulated HUVECs on fibronectin instead of collagen. The authors should comment on the reasons why they optimized this different protocol in order to help the readers understand what are the most relevant variables that should be taken into account when optimizing this protocol in the lab.
- Reference 18, line 222 is incorrect. Please check that all references are numbered properly.
Reviewer 3 Report
In this article, the authors demonstrate the utility of a microfluidic device to study the role of the Neutrophil in immunothrombosis. They demonstrate convincingly that the basic parameters of flow and thrombosis model that of mammalian vessels. Where the model falls short is looking at arterial sheer, where the authors neglect to evaluate if the endothelial cells are releasing vWF and how that effects neutrophil adhesion.
It is recommended that at least vWF if not both vWF and ADAMS 13 be evaluated in their system before publication.
